# Short-Term Consumption of Hydrogen-Rich Water Enhances Power Performance and Heart Rate Recovery in Dragon Boat Athletes: Evidence from a Pilot Study

**DOI:** 10.3390/ijerph19095413

**Published:** 2022-04-29

**Authors:** Gengxin Dong, Jiahui Fu, Dapeng Bao, Junhong Zhou

**Affiliations:** 1School of Sport Medicine and Physical Therapy, Beijing Sport University, Beijing 100084, China; ddgx0419@163.com; 2School of Strength and Conditioning Training, Beijing Sport University, Beijing 100084, China; 2020240650@bsu.edu.cn; 3China Institute of Sport and Health Science, Beijing Sport University, Beijing 100084, China; 4Hebrew Senior Life Hinda and Arthur Marcus Institute for Aging Research, Harvard Medical School, Boston, MA 02131, USA; junhongzhou@hsl.harvard.edu

**Keywords:** hydrogen-rich water, power, heart rate, recovery, dragon boat

## Abstract

(1) Background: Exercise that exceeds the body’s accustomed load can lead to oxidative stress and increased fatigue during intense training or competition, resulting in decreased athletic performance and an increased risk of injury, and the new medicinal H_2_ may be beneficial as an antioxidant. Therefore, we explored the effect of short-term supplementation of hydrogen-rich water (HRW) on the work performance and fatigue recovery of dragon boat athletes after training. (2) Methods: Eighteen dragon boat athletes who trained for 4 h a day (2 h in the morning and 2 h in the afternoon) were divided into an HRW group (n = 9) and a placebo water (PW) group (n = 9), drinking HRW or PW for 7 days. Each participant completed 30 s rowing dynamometer tests, monitoring the heart rate at baseline (i.e., Day 1) and after the intervention (on Day 8). (3) Result: Drinking HRW increased the maximum power and average power of the 30 s rowing test and decreased the maximum heart rate during the period. After the rowing test, the HRW group’s heart rate dropped significantly after 2 min of recovery, while the PW group’s heart rate did not drop. There was no significant difference between the 30 s rowing distance and the predicted duration of rowing 500 m. (4) Conclusions: Drinking HRW in the short term can effectively improve the power performance of dragon boat athletes and is conducive to the recovery of the heart rate after exercise, indicating that HRW may be a suitable means of hydration for athletes.

## 1. Introduction

Dragon boating is one type of exercise with a high intensity of physical load, inducing high physical fatigue. The fatigue oftentimes alters the regulation of human physiological systems by disrupting the redox balance [1,2] and disturbing acid–base homeostasis in these systems [3]. The oxidative stress and acidification due to fatigue from such high-intensity exercise also negatively impact athletic performance [4,5] and slow post-exercise recovery in athletes [3]. Therefore, accelerating recovery from fatigue after rowing exercises is critical to maintain the intact function in athletes, helping to reduce the incidence of overtraining and injuries and ultimately improve rowing performance [6].

When performing low-intensity exercise, reactive oxygen species (ROS) and reactive nitrogen species (RNS) are generated at a low rate and subsequently scavenged by the antioxidant system. However, the performance of high-intensity exercise can lead to an increase in the production of ROS and RNS. When the antioxidant defense capacity is also exceeded, the oxidative stress state appears [2], which is closely related to muscle fatigue [7,8], leading to decreased physical function, muscle damage and inflammation, and delayed-onset muscle soreness [9,10]. Meanwhile, high-intensity exercise also induces metabolic acidosis, increases blood and muscle pH [11], and thus affects exercise performance and fatigue recovery. Studies have shown that supplementation with antioxidants can reduce the adverse effects of oxidative stress on exercise performance during exercise [12,13], and supplementation with alkalizing agents can reduce metabolic acidosis [14]. However, conventional antioxidants scavenge those ROS that are beneficial to health [15,16], and the alkalizers have significant toxicity accompanied by side effects (e.g., stomach pain, diarrhea, nausea, vomiting) [17]. Therefore, it is highly demanded to develop novel types of supplements that can safely help reduce oxidative stress and alleviate acidosis to accelerate recovery after high-intensity exercise (e.g., dragon boating), without altering the ROS signaling without side effects.

Since Ohsawa et al. first reported that hydrogen has powerful selective antioxidant properties [18], the biological effects of hydrogen have been widely studied. Hydrogen-rich water (HRW) is one such safe nutrient that can be used as an alkalizing agent [19] and antioxidant [20,21], potentially helping to accelerate post-exercise recovery. Studies have shown the benefits of using HRW for reducing blood lactate levels [22,23], increasing blood pH [24], inhibiting exercise-induced oxidative stress [25], and thus alleviating exercise-induced muscle fatigue [22]. However, the effects of taking HRW on athletic performance are uncertain [24,26], and only immediate effects of taking one dose of HRW have been explored; the benefits of taking HRW through a period of time on the performance of long-term high-intensity exercise (e.g., dragon boating) are still unknown.

Therefore, in this randomized, single-blinded pilot study, we examined the effects of taking HRW for 7 days on the rowing performance and post-exercise recovery in a group of elite dragon boat athletes. We hypothesized that, compared to those who take purified water (i.e., controls), participants who take HRW would perform the rowing task better, and their recovery would also be better, and that HRW can be used as a convenient and effective anti-fatigue hydration strategy.

## 2. Materials and Methods

### 2.1. Subjects

Eighteen dragon boat athletes were randomly assigned into a hydrogen-rich water (HRW) group (23.22 ± 1.09 years, n = 9) and a placebo water (PW) group (22.67 ± 0.87 years, n = 9). Each group included 6 male athletes and 3 female athletes; there was also no significant difference in baseline values between the two groups when divided by gender (Table 1). Participants were well-experienced athletes (professional dragon boat experience of ~3 years) who train approximately 28 h per week and were free from any dietary supplements for the past 6 months. All participants were in excellent health, without cardiovascular, respiratory, or endocrine system diseases, as the preliminary medical screening confirmed. None of the subjects smoked or took any supplements/drugs during the experiment. The baseline characteristics of the participants are shown in Table 1. Before the start of the experiment, all subjects were informed of the relevant benefits and possible risks involved in participating in this study and provided signed informed consent as approved by the Ethics Committee of Exercise Science Experiment of Beijing Sport University.

### 2.2. Hydrogen-Rich Water

The HRW was prepared using an electrolysis device (Zhiheng Hydrogen Health Technology Co., Ltd., Fuzhou, China) with a transparent Tritan^TM^ tank body and an electrolysis generator. The concentration of H_2_ in the HRW was 1600 ppb. To enable the blinding, the outer packages of the hydrogen and placebo water were the same so that subjects were not aware of which water they were drinking.

### 2.3. Protocol

Athletes performed 4 h of daily water training (2 h in the morning and 2 h in the afternoon) for 8 days, with an average daily rowing distance of not less than 3000 m. The subjects ate meals that were the same as those they usually ate in everyday life during the experiment and were asked to not take any dietary supplements and drugs. Only 500 mL of purified water was added during the afternoon training. From Day 2 to Day 8, subjects were asked to drink one bottle immediately after the morning training and one bottle immediately after the afternoon training. Each subject thus ingested 1000 mL of HRW or PW per day.

Immediately after the afternoon training on Day 1 and Day 8, each subject performed a 30 s rowing test. The researchers recorded the maximum power, average power, stroke distance, and predicted time taken to stroke 500 m at the average speed. The Firstbeat heart rate belt was used to monitor the resting heart rate, the maximum heart rate while performing the task, the heart rate immediately after the test, the heart rate after 1 min of recovery, the heart rate after 2 min of recovery, and the heart rate after 3 min of recovery.

### 2.4. Rowing Dynamometer 30 s Rowing Test

All subjects performed a 30 s full-strength rowing test on a rowing dynamometer (Model D, Concept II Inc., Morristown, VT, USA) after the afternoon training on Day 1 and Day 8. The subjects first performed a 3 min warm-up exercise on a rowing dynamometer (slow speed, with a drag coefficient of 2), followed by a 30 s full-strength exercise on the dynamometer (with a drag coefficient of 5). During the full-strength exercise, the subjects chose their own physical strength distribution method (pacing strategy) and were verbally encouraged by the testers to perform the longer distance for as long as possible within the specified time (the dynamometer can be displayed).

### 2.5. Heart Rate Monitoring

We used the Firstbeat heart rate belt (Firstbeat Analytics, Jyvaskyla, Finland) to monitor the heart rate changes of the athletes during the test. The heart rate belt was worn on the athlete’s chest, and the receiver was positioned to the left of the midline. The elastic band was adjusted so that the position of the receiver did not change during the athlete’s exercise. All subjects wore the heart rate belt before training on Day 1 and Day 8, and their heart rate was recoded after resting for 5 min. After the water training in the afternoon, the participants wore the Firstbeat heart rate belt again for the rowing dynamometer test and rested for 3 min after the test. The researchers monitored the tablet and recorded the subjects’ maximum heart rate during the rowing dynamometer test, the heart rate immediately after the test, the heart rate after 1 min of recovery, the heart rate after 2 min of recovery, and the heart rate after 3 min of recovery. The percent change from baseline in heart rate at each recovery time was then calculated (e.g., heart rate after 1 min of recovery − resting heart rate)/resting heart rate) for each subject and used to characterize the recovery of the heart rate. The smaller the change, the better the recovery.

### 2.6. Statistical Analysis

Statistical analyses were performed using SPSS 25.0 (IBM, Chicago, IL, USA). The significance level was set at *p* < 0.05. Descriptive statistics (i.e., mean, standard deviation (SD)) were used to summarize the demographic characteristics of the participants and study outcomes. Shapiro–Wilk tests were used to examine if the data were normally distributed. Independent sample *t*-tests were used to examine the demographic characteristics of the participants and pre-intervention test results (baseline values). Two-way (group × time) mixed design analyses of variance (ANOVAs) were used to examine the effects of HRW on those outcomes. The independent factor was group (HRW group, PW group), and the repeated measures factor was time (before intervention, after intervention); their interaction effects were examined. Post hoc analyses were performed if there were significant interactions. Secondarily, one-way repeated measures ANOVA was used to examine the effects of the intervention on outcomes within each group, and one-way ANOVA was used to examine the differences between groups after the intervention.

## 3. Results

### 3.1. Rowing Dynamometer 30 s Rowing Test Results

Eighteen participants completed all study tests, and their data were included in the analysis. There were no significant differences in the demographic characteristics of the participants and pre-intervention outcomes (Table 1). Two-way mixed design ANOVA models showed a trend towards a significant interaction in the maximum power of the 30 s rowing test (F = 3.259, *p* = 0.090), while no significant interaction in other outcomes of the rowing test was observed (i.e., the average power (F = 0.571, *p* = 0.461), the distance (F = 0.330, *p* = 0.573), and the predicted time of rowing 500 m (F = 0.272, *p* = 0.609)) (Table 2). However, in the secondary analyses, the one-way repeated measures ANOVA model showed that within the HRW group, the maximum power (F = 9.396, *p* = 0.015) and average power (F = 5.544, *p* = 0.046) were significantly improved after the intervention as compared to baseline, while no significant changes in these two outcomes within the PW group were observed (F < 0.093, *p* > 0.768). No significant changes in the distance of the 30 s rowing test and the predicted time of rowing 500 m were observed within either the HRW group (distance: F = 3.002, *p* = 0.121; predicted time: F = 3.513, *p* = 0.098) or the PW group (distance: F = 2.057, *p* = 0.189; predicted time: F = 2.212, *p* = 0.175). One-way ANOVA showed no significant difference between the HRW and PW groups in each outcome of the rowing test after the intervention.

### 3.2. Heart Rate Monitoring Results

#### 3.2.1. Maximum Heart Rate Monitoring Results during Rowing Test

Two-way mixed design ANOVA models showed a significant interaction in the maximum heart rate during the rowing test (F = 5.741, *p* = 0.029) (Table 3). Post hoc analyses revealed that the maximum heart rate during the rowing test after the intervention was not significantly different between the HRW and PW groups (F = 0.055, *p* = 0.818), and that the maximum heart rate was significantly decreased in the HRW group after the intervention (F = 8.391, *p* = 0.020), but no such significant change was observed in the PW group (F = 0.123, *p* = 0.735).

#### 3.2.2. Heart Rate Recovery Monitoring Results after Rowing Test

There were no significant differences in heart rate changes between the two groups before the intervention (Table 1). We used the percent change from baseline in heart rate, that is (monitoring heart rate—resting heart rate)/resting heart rate, to represent the recovery of the heart rate. Two-way mixed design ANOVA models showed no significant interaction in the resting heart rate (F = 1.370, *p* = 0.259), the percent change from baseline in heart rate immediately after the rowing test (F = 4.311, *p* = 0.054), the percent change from baseline in heart rate after 1 min of recovery (F = 4.247, *p* = 0.055), the percent change from baseline in heart rate after 2 min of recovery (F = 3.621, *p* = 0.075), and the percent change from baseline in heart rate after 3 min of recovery (F = 4.090, *p* = 0.060). However, in the secondary analyses, the one-way repeated measures ANOVA model showed that within the HRW group, the percent change from baseline in heart rate after 2 min of recovery (F = 14.124, *p* = 0.006) and the percent change from baseline in heart rate after 3 min of recovery (F = 17.204, *p* = 0.003) were significantly decreased after the intervention as compared to baseline, while no significant changes in these two outcomes within the PW group were observed (F < 2.233, *p* > 0.173). No significant changes in the resting heart rate, the percent change from baseline in heart rate immediately after the rowing test, and the percent change from baseline in heart rate after 1 min of recovery were observed within either the HRW group (resting heart rate: F = 0.527, *p* = 0.489; growth rate after 0 min: F = 3.040, *p* = 0.119; growth rate after 1 min: F = 3.225, *p* = 0.110) or the PW group (resting heart rate: F = 1.118, *p* = 0.321; growth rate after 0 min: F = 1.303, *p* = 0.287; growth rate after 1 min: F = 1.076, *p* = 0.330). One-way ANOVA showed no significant difference between the HRW and PW groups in the resting heart rate (F = 1.174, *p* = 0.295), the percent change from baseline in heart rate immediately after the rowing test (F = 0.943, *p* = 0.346), and the percent change from baseline in heart rate after 1 min of recovery after the intervention (F = 0.355, *p* = 0.559), but the percent change from baseline in heart rate after 2 min of recovery of the HRW group was significantly lower than that of the PW group after the intervention (F = 4.829, *p* = 0.043), and the percent change from baseline in heart rate after 3 min of recovery of the HRW group tended to be significantly lower than that of the PW group after the intervention (F = 4.486, *p* = 0.050) (Table 4). In addition, we compared changes in heart rate within groups before and after the intervention using one-way repeated measures ANOVA. The results show that within the HRW group, the heart rate of the subjects was significantly decreased 2 min after the rowing test, while such a significant decrease was not observed within the PW group (Figure 1).

## 4. Discussion

In this pilot study, we observed, for the first time, that consuming HRW for a period of time (i.e., 7 days) after prolonged high-intensity training is of promise to improve power performance and help the recovery of the heart rate in dragon boaters. Greater within-group changes in the maximum and average power of the rowing test were observed in the HRW group as compared to the PW group, indicating the potential benefits of HRW for sprint performance after prolonged exercise. Compared to the PW group, the HRW group had greater within-group changes in the maximum heart rate of the rowing test and the heart rate recovery rate 2–3 min after the test, indicating accelerated recovery from the high-intensity exercise.

We observed that subjects in the HRW group had a significantly greater increase in the maximum and average power of the 30 s full-strength rowing test and a decrease in the maximum heart rate and faster recovery of the heart rate after the test as compared to the PW group. These results may suggest that HRW can help reduce oxidative stress and metabolic acidosis caused by high-intensity exercise. Studies have demonstrated the relationship between oxidative stress and physical fatigue [27,28] and explained the mechanism of free radical-induced fatigue at the cellular and molecular levels [29,30], including various hypotheses that ROS affect fatigue, such as the hypotheses that high levels of ROS alter mitochondrial function [31,32], increased utilization of anaerobic pathways leads to elevated inorganic phosphate (Pi) levels and acidosis [31], an altered redox status leads to muscle contraction (contractile proteins) and muscle contraction (calcium pump) changes in controls [33], and action potentials alter muscle contraction [34]. HRW can act as an antioxidant to selectively reduce harmful ROS in the body while acting as an alkalizing agent to reduce the pH of the blood and muscles [24], thus reducing physical fatigue. Compared to low-intensity exercise, prolonged high-intensity training oftentimes leads to a large increase in free radicals in the human biophysiological system, exceeding the limit of the ability to self-eliminate such free radicals. In this situation, the supplemented antioxidants (e.g., those from HRW) can help reduce physical fatigue and thus eliminate the damage caused by free radicals, helping the maintenance of sports state and performance. Therefore, HRW may be particularly helpful for maintaining athletic performance and accelerating the recovery from fatigue after high-intensity exercise.

However, the benefits of HRW we observed here are still limited (e.g., no effects on rowing distance and the predicted rowing time for 500 m at average speed). One of the most significant limitations is the small sample size (n = 9 in each group) and short duration of the test (only 30 s). In addition, although we controlled for the athletes’ diet and energy intake during the experimental period, the athletes’ daily diet may still have a potential impact on exploring the effect of hydrogen-rich water, which should also be carefully controlled and assessed in future studies. Future studies with a larger sample size and implementing longer-term exercises/tests are thus highly demanded to examine and confirm the observations in this study and the effects of HRW on sport performance. Additionally, we only measured the functional performance in this study, and the underlying pathway through which HRW influences recovery and functional performance needs to be explicitly examined in future studies with biophysiological assessments, which will ultimately provide critical knowledge for the optimization of HRW-based strategies (e.g., dose–response relationship of HRW with performance) for accelerating recovery and maintaining performance in athletes.

## 5. Conclusions

This pilot study demonstrated that HRW is of great promise to help accelerate the recovery from high-intensity exercise and to benefit the functional performance of rowing in elite dragon boat athletes, which is beneficial for them to increase power output, gain an advantage in the final sprint stage, and reduce physical damage from accumulated fatigue. Although further studies are absolutely warranted, drinking HRW may be a beneficial hydration strategy for athletes’ recovery.

## Figures and Tables

**Figure 1 ijerph-19-05413-f001:**
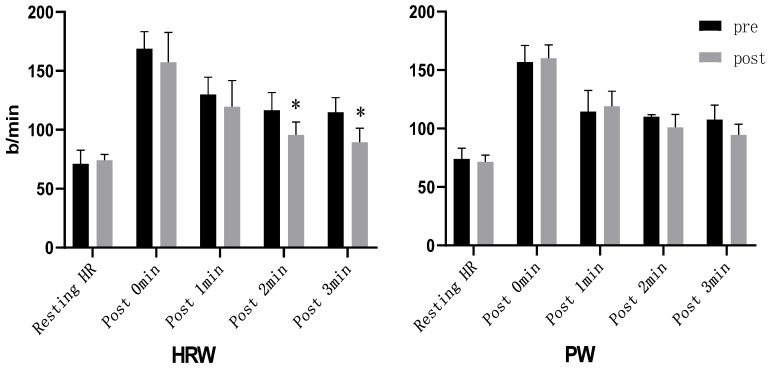
Heart rate changes after the rowing test. * Indicates that there is a statistically significant difference between the groups after the intervention and before the intervention, *p* < 0.05.

**Table 1 ijerph-19-05413-t001:** Baseline characteristics of participants.

Characteristics	Grouped by Gender				Total		
	Gender	HRW	PW	*p*-Value	HRW (n = 9)	PW (n = 9)	*p*-Value
Age (years)	M (n = 6)	22.67 ± 0.82	22.67 ± 0.52	1.000	23.22 ± 1.09	22.67 ± 0.87	0.249
	F (n = 3)	24.33 ± 0.58	22.67 ± 1.53	0.152			
Height (cm)	M (n = 6)	174.17 ± 4.22	173.00 ± 6.07	0.707	170.11 ± 7.27	168.67 ± 8.84	0.710
	F (n = 3)	162.00 ± 4.36	160.00 ± 7.21	0.702			
Body weight (kg)	M (n = 6)	66.50 ± 6.09	67.17 ± 7.94	0.874	64.89 ± 6.09	61.94 ± 11.56	0.509
	F (n = 3)	61.67 ± 5.69	51.50 ± 11.46	0.241			
MP (w)	M (n = 6)	462.00 ± 42.21	487.50 ± 144.85	0.688	401.00 ± 111.38	390.22 ± 189.97	0.885
	F (n = 3)	279.00 ± 108.06	195.67 ± 82.00	0.347			
AP (w)	M (n = 6)	351.83 ± 31.00	365.00 ± 127.53	0.811	300.89 ± 91.08	290.78 ± 153.25	0.867
	F (n = 3)	199.00 ± 86.13	142.33 ± 60.93	0.405			
30 s rowing distance (m)	M (n = 6)	179.17 ± 5.34	178.83 ± 19.25	0.968	172.22 ± 18.36	163.44 ± 29.33	0.458
	F (n = 3)	158.33 ± 29.02	132.67 ± 19.55	0.273			
Predicted time of rowing 500 m (s)	M (n = 6)	99.68 ± 3.12	100.97 ± 8.26	0.729	107.68 ± 14.92	113.42 ± 22.18	0.528
	F (n = 3)	123.67 ± 17.04	138.33 ± 20.03	0.389			
MHR (b/min)	M (n = 6)	178.17 ± 12.64	169.83 ± 13.67	0.299	176.89 ± 11.36	162.78 ± 17.22	0.057
	F (n = 3)	174.33 ± 10.12	148.67 ± 16.44	0.083			
RHR (b/min)	M (n = 6)	70.50 ± 6.22	72.50 ± 9.38	0.673	71.33 ± 11.20	74.33 ± 8.85	0.537
	F (n = 3)	73.00 ± 19.97	78.00 ± 7.94	0.708			
HRPC immediately after the test (%)	M (n = 6)	147.57 ± 40.24	123.68 ± 42.52	0.341	142.73 ± 39.90	114.47 ± 36.51	0.137
	F (n = 3)	133.03 ± 45.91	96.07 ± 7.01	0.240			
HRPC after 1 min of recovery (%)	M (n = 6)	88.33 ± 34.13	72.08 ± 32.69	0.419	85.59 ± 32.48	56.88 ± 34.65	0.089
	F (n = 3)	80.10 ± 35.22	26.48 ± 7.34	0.061			
HRPC after 2 min of recovery (%)	M (n = 6)	73.65 ± 32.21	63.60 ± 30.64	0.592	66.26 ± 30.15	50.21 ± 32.52	0.293
	F (n = 3)	51.50 ± 23.47	23.42 ± 16.41	0.165			
HRPC after 3 min of recovery (%)	M (n = 6)	67.06 ± 31.17	56.77 ± 24.63	0.540	64.01 ± 26.92	46.66 ± 25.05	0.176
	F (n = 3)	57.92 ± 19.65	26.45 ± 8.63	0.064			

Note: HRW, hydrogen-rich water; PW, placebo water; M, male; F, female; MP, maximum power; AP, average power; MHR, maximum heart rate; RHR, resting heart rate; HRPC, heart rate percent change.

**Table 2 ijerph-19-05413-t002:** Performance in 30 s rowing test before and after the intervention within the HRW and PW groups.

Index	Group	Pre	Post	Interaction *p*-Value
MP (w)	HRW	401.00 ± 111.38	442.67 ± 112.47 *	0.090
	PW	390.22 ± 189.97	390.11 ± 155.14	
AP (w)	HRW	300.89 ± 91.08	321.33 ± 77.47 *	0.461
	PW	290.78 ± 153.25	296.22 ± 123.52	
30 s rowing distance (m)	HRW	172.22 ± 18.36	177.00 ± 14.54	0.573
	PW	163.44 ± 29.33	172.00 ± 28.28	
Predicted time of rowing 500 m (s)	HRW	107.68 ± 14.92	104.39 ± 10.21	0.609
	PW	113.42 ± 22.18	111.31 ± 19.56	

* Indicates that the one-way repeated measures ANOVA showed significant differences between groups before and after the intervention, *p* < 0.05; HRW, hydrogen-rich water; PW: placebo water; MP, maximum power; AP, average power.

**Table 3 ijerph-19-05413-t003:** Maximum heart rate before and after the intervention within the HRW and PW groups.

Group	Pre (b/min)	Post (b/min)	Interaction *p*-Value
HRW	176.89 ± 11.36	162.44 ± 21.39 *	0.029
PW	162.78 ± 17.22	164.33 ± 11.31	

* Indicates that the one-way repeated measures ANOVA showed significant differences between groups before and after the intervention, *p* < 0.05; HRW: hydrogen-rich water; PW: placebo water.

**Table 4 ijerph-19-05413-t004:** Heart rate recovery before and after the intervention in the HRW and PW groups.

Group		Resting Heart Rate (b/min)	HRPC Immediately after the Test (%)	HRPC after 1 min of Recovery (%)	HRPC after 2 min of Recovery (%)	HRPC after 3 min of Recovery (%)
HRW	Pre	71.33 ± 11.20	142.73 ± 39.90	85.59 ± 32.48	66.26 ± 30.15	64.01 ± 26.92
	Post	74.33 ± 4.69	111.62 ± 30.94	60.70 ± 25.31	28.68 ± 12.70 *	20.47 ± 14.77 *
PW	Pre	74.33 ± 8.85	114.47 ± 36.51	56.88 ± 34.65	50.21 ± 32.52	46.66 ± 25.05
	Post	71.67 ± 5.70	125.42 ± 29.34	66.79 ± 17.20	41.58 ± 12.20 ^#^	32.10 ± 7.27

* Indicates that the one-way repeated measures ANOVA showed significant differences between groups before and after the intervention, *p* < 0.05; ^#^ indicates that there is a statistical difference between the HRW and PW groups at the same time, *p* < 0.05; HRW: hydrogen-rich water; PW: placebo water; HRPC: heart rate percent change.

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
