# Peer review of "Short-Term Consumption of Hydrogen-Rich Water Enhances Power Performance and Heart Rate Recovery in Dragon Boat Athletes: Evidence from a Pilot Study"

_ijerph, 2022, doi:10.3390/ijerph19095413_

Round 1

Reviewer 1 Report

Dear authors,

From my point of view, this manuscript is not in a form to be published in the IJERPH.  The sample size is too small for meaningful extrapolation, and ideally, it should have examined a larger sample that produced more generalizable findings despite it being a pilot study. There is a lack of information about the energy, nutrient, and liquid intake of the participants as well as the mean intensity training load in the training group sessions. Both would have to be considered key factors in your study. Furthermore, I do not understand why the groups are mixed-sex when you are talking about power performance in dragón boat athletes.

Finally, here I propose some recommendations and mistakes detected to improve your manuscript:

Title:

I would add the description of this is a pilot study

Introduction:

Please, add a scientific cite to justify the statement at the end of the sentences of lines 50 and 52. And also, modify the position of parentheses in the sentence of line 54.

Materials and Methods:

There is a lack of information about the energy, nutrient, and liquid intake as well as the mean intensity training load in the training group sessions, which should be considered in the study.

Please, unify spaces and the use of the International abbreviations in table 1.

Moreover, please, add the material information of the Firstbeat heart rate belt from lines 118 to 104.

Results:

I suggest adding the International abbreviations in the correct format in table 2.

Discussion:

From my point of view, the most significant limitations are not considered in this section.

Yours faithfully,

Author Response

请参阅附件。

Reviewer 2 Report

This manuscript, on a prospective study, was judged that the content of the manuscript is generally okay.

However, I decided it as “minor revision” because the author used a criterion called "heart rate growth rate" and I thought this indicator was not common. I would like the author to explain this index a little more.

Also, in Table 4, I was wondering if whether there was any difference in each time point between HRW and PW groups in heart rate recovery before intervention would affect the results, so I made a comment.

Round 2

Reviewer 1 Report

Dear authors,

I consider that all my comments and suggestions had been answered point-by-point.
In any case, I still detect some grammar mistakes in your manuscript such as the term “gende”  for “gender” in table 1, the use of the international metric units (SI) without capital letters, or I cannot understand why the use of the acronym “HRPC”: (heart rate percent change) is not used it in the text of your manuscript in some cases (i.e. the percent change from baseline in heart rate) to facilitate understanding to the readers and unify the terms used in the manuscript.

Despite my comments and after correcting or considering them, I accept the manuscript to be published before the final decision of the IJERPH Editor.

Yours faithfully,